# Implant Periapical Lesion: Clinical and Histological Analysis of Two Case Reports Carried Out with Two Different Approaches

**DOI:** 10.3390/bioengineering9040145

**Published:** 2022-03-29

**Authors:** Roberto Luongo, Fabio Faustini, Alessandro Vantaggiato, Giuseppe Bianco, Tonino Traini, Antonio Scarano, Eugenio Pedullà, Calogero Bugea

**Affiliations:** 1Independent Researcher, 70121 Bari, Italy; rl66@nyu.edu; 2Independent Researcher, 29020 Piacenza, Italy; info@dentalfaustini.it; 3Independent Researcher, 73100 Lecce, Italy; alessandrovantaggiato@gmail.com; 4Department of Medical, Oral and Biotechnological Sciences, University of “G. D’Annunzio” of Chieti-Pescara, 66100 Chieti, Italy; gb373@nyu.edu (G.B.); ascarano@unich.it (A.S.); 5Department of Innovative Technologies in Medicine & Dentistry, University of “G. D’Annunzio” of Chieti-Pescara, 66100 Chieti, Italy; t.traini@unich.it; 6Department of General Surgery and Surgical-Medical Specialties, University of Catania, 95100 Catania, Italy; eugenio.pedulla@unict.it

**Keywords:** implant periapical lesion, implant failure, peri-implantitis, endodontic surgery, complication

## Abstract

Periapical implantitis (IPL) is an increasingly frequent complication of dental implants. The causes of this condition are not yet entirely clear, although a bacterial component is certainly part of the etiology. In this case series study, two approaches will be described: because of persistent IPL symptoms, a patient had the implant removed and underwent histological analysis after week 6 from implantation. The histomorphometric examination revealed a 35% bone-implant contact area involving the coronal two-thirds of the implant. The apical portion of the fixture on the other hand was affected by an inflammatory process detectable on radiography as a radiolucent area. The presence of a probable root fragment, detectable as an imprecise radiopaque mass in the zone where the implant was later placed, confirms the probable bacterial etiology of this case of IPL. On the other hand, in case number 2, the presence of IPL around the fixture was solved by surgically removing the implant apical third as well as the adjacent tooth apex. It may be concluded from our histological examination that removal of the apical portion of the fixture should be considered an effective treatment for IPL since the remaining implant segment remains optimally osseointegrated and capable of continuing its function as a prosthetic abutment. Careful attention, however, is required at the implantation planning stage to identify in advance any sources of infection in the edentulous area of interest which might compromise the final outcome.

## 1. Introduction

Although the predictability of endosseous implants is well supported in studies, the possibility of failure in the long and short term still exists [1]. Implant failures have been defined as a host tissue inadequacy in stabilizing or maintaining osseointegration. Correlated with the time of onset, such failures may be classified as early or late depending on whether they occur before or after occlusal loading. Implant failures may be due to iatrogenic causes associated with a less-than-optimal surgical technique, bacterial causes secondary to contamination of the implant site during or after insertion of the fixture, possible systemic comorbidities, and excessive occlusal loading [1,2,3,4,5].

The clinical signs of early failure of an implant are local inflammation of the peri-implant hard and soft tissue, defined above as “peri-implantitis”, and sometimes accompanied by secretion of purulent exudate, bleeding, and probing depth more than 3 mm. Histological findings from the peri-implant area include the presence of an inflammatory cellular infiltrate, epithelial proliferation, bacteria, and ultimately areas of osseous necrosis [1]. Radiographically, however, a radiolucent zone is detectable in the peri-implantitis affected area around the fixture, resembling an osseous crater at the crestal level that extends in the apical direction. Peri-implantitis and periodontitis may trigger interactions between host immune defense mechanism and bacteria that eventually lead to implant failures.

The most common type of peri-implantitis generally involves the more coronal part of the implant and only later tends to spread in the apical direction, sometimes leaving the apical portion of the fixture still firmly integrated in the crestal bone [6].

It may happen occasionally that peri-implantitis will develop apically in the same manner as a periapical lesion of a dental structure, i.e., without involvement of the coronal crest bone. This particular condition has been defined as implant periapical lesion (IPL), apical peri-implantitis, retrograde peri-implantitis, or endodontic implant pathology. It should be considered a distinct form relative to the more common form of peri-implantitis, which involves the coronal portion of the fixture [7,8].

A retrospective study of approximately 3800 implants found an IPL incidence of 0.26%. In another study, Quirynen reported an IPL incidence of 1.6% in the maxilla and 2.7% in the mandible [9,10].

The best evidence-supported etiology of IPL is diffusion of pathogenic bacteria from infected dental remnants present in the bone around the tip of the implant, or implant-adjacent dental structures with endodontic periapical lesions [9,10]. Depending on the sources releasing the peri-implant infection, the IPL may be divided into Type 1, when diffusion proceeds from the fixture to the adjacent dental structure and Type 2, when the structure is affected first by an inflammatory process diffusing to the nearby implant [11].

This article describes two cases of IPL, which were treated with two different approaches: a surgical removal of the entire affected fixture and a surgical removal of just the apical third of the implant as well as the adjacent tooth apex. 

## 2. Case Report n.1 

A 45-year-old patient in good general health, non-smoker, with partial edentulism in the right mandible, presented to our service for rehabilitation of the missing structures with implant-supported prostheses.

On presentation, the patient was already edentulous in the area of the right mandibular second premolar and the first and second molar for more than 10 years. Before developing an adequate treatment plan, we ordered a pantomograph (Figure 1), after which CT was needed because of the close proximity of the alveolar canal.

The CT scan demonstrated the presence of a small area of radiopacity of the bony structure at teeth 46–47, the nature of which could not be defined (Figure 2). The patient treatment plan proposed rehabilitation with a 3-unit fixed partial denture supported on 2 implants at #45 and #47. The patient was administered antibiotic therapy from day 1 of the intervention: the regimen was 1 g amoxicillin/clavulanic acid (Augmentin; Glaxo SmithKline, Verona, Italy), 1 g q12 h for 6 days postoperatively for a total of 7 days of treatment. Following local/regional anesthesia with mepivacaine 2% and adrenaline 1:100,000 (Scandonest, Septodont, Saint-Maur des Fossés, France), a full-thickness flap was raised from tooth 44 to 47. Trunk anesthesia was not performed so as to preserve the sensitivity of the alveolar nerve during the subsequent osteotomies, thereby avoiding injury of the vascular-neural bundle. Later, two osteotomies were prepared free-handed at teeth 45 and 47 with the aid of a surgical template. Lastly, two cylindrical Biomet 3i (Palm Beach Gardens, FL, USA) implants were placed, one with 4 mm diameter × 10 mm length at #45 and the other of the expanded platform (Xp) type, i.e., an implant diameter of 4 mm and a platform with 5 mm diameter × 10 mm length at #47. In view of the high primary stability obtained, it was decided to place the healing screws directly on the fixtures according to the procedure for one-stage implants (Figure 3A). Lastly, 4/0 interrupted silk sutures were placed for flap closure. On completion of the surgical procedure, follow-up intraoral radiography was performed to confirm correct implant placement.

Several hours after the intervention and on subsequent days, the patient spontaneously reported continuous pain of a pulsating type, similar to pulpits pain, close to the right mandibular molar area. The authors made the diagnosis of IPL after intraoral radiographic examination. The IPL was relieved only through administration of an analgesic (nimesulide 100 mg—Aulin, Roche, Milan, Italy), but no neurosensory changes were reported for the right mandible.

The IPL symptoms lasted for more than a month from the date of the intervention, during which intraoral radiographs were taken to detect possible abnormalities in loading of the peri-implant hard tissue. The implant at tooth 47 did not present any bleeding or probing depth more than 3 mm, but an intraoral radiograph showed an area of radiolucency close to the apical segment of the fixture (Figure 3B). Because of the persistence of IPL symptoms, it was decided (with the patient’s consent) on week 6 after the implantation to remove the implant at tooth 47 (Figure 3C). After antibiotic therapy with 1 g Augmentin twice daily, starting on preoperative day 1 and continuing for 6 days afterwards, the area was infiltrated with local/regional anesthesia using mepivacaine 2% and adrenaline 1:100,000 and we prepared a muco-periosteal flap to expose the crest at teeth 46–47. Although abnormal loading of the peri-implant crestal bone was not found (Figure 3D), we drilled the implant with a 6 mm diameter trephine bur at #47 (Figure 3E,F). Then, we placed another implant at #46 (Biomet 3i, Palm Beach Gardens, FL, USA, 4 mm diameter, 10 mm length), free-handed using a one-stage procedure which required the immediate insertion of a healing screw on the same day. Lastly, we removed the granulation tissue present and packed the drilling site with a collagen sponge (Hemocollagene, Septodont, Saint-Maur des Fossés, France). The flaps were sutured with 4/0 silk interrupted sutures (Figure 4A–C). After the intervention, the patient did not report IPL symptoms, nor were there any signs of local inflammation. 

After approximately three months from the intervention, in which no complications were observed, we proceeded to mount prostheses on the implants at teeth 45 and 46, which consisted in a two-unit fixed partial denture supported on 2 implants at #45 and #46 (Figure 4D).

### 2.1. Histological Preparation 

The drilled implant was preserved in buffered formalin 10%, dehydrated in a progressive series of washings with alcohol, and embedded in glycol methacrylic resin in preparation for histological analysis (Technovit 7200 VCL, Kulzer, Wehreim, Germany). After polymerization, the implant was sectioned along its longitudinal axis with a high-speed diamond disk (Precise System, Assing, Rome, Italy). Sections of approximately 46 µm thickness were obtained and then stained with basic fuchsin and toluidine blue. A second staining procedure was performed with von Kossa and basic fuchsin to evaluate the level of bone mineralization.

### 2.2. Histological Analysis

The histological analysis shows adequate osseointegration in the coronal aspect, with a BIC of approximately 35%, while the apical portion of the implant is totally free of bone. (Figure 5).

## 3. Case Report n. 2 

In September 2020, a 52-year-old patient came to our attention with pain and swelling in the upper right premolar area. The fixture showed no clinical signs of peri-implantitis and no mobility but the whole area was sensitive upon percussion. Upon radiographic examination, a radiolucent area involving the implant apical third as well as the root apex of tooth number 5 was observed (Figure 6A). The root canal treatment seemed to be correctly performed as the obturation material was thoroughly compacted, lateral accessory canals were present and filled. The tooth was restored with a metallic post and core and a prosthetic crown. A surgical approach was chosen to remove the infected apexes of both the fixture and tooth number 5.

Antibiotic therapy was administered: 1 g amoxicillin/clavulanic acid (Augmentin; GlaxoSmithKline, Verona, Italy) every 12 h for 6 days.

A para-marginal incision was performed and a full thickness flap raised from tooth number 6 up to tooth number 3 with two vertical relieving incisions (Figure 6B). Access to the lesion was created by means of the bur H255E.314.012 (Komet, Germany). A horizontal cut was made with the same bur both on the root and on the fixture apex and all the granulation tissue in the surrounding bone was scraped off (Figure 6C).

The retropreparation of tooth 5 was done with surgical retrotip (R1D, Piezomed, W&H, Bürmoos, Austria) and it was filled with Biodentine (Septodont, Saint-Maudefuse, France).

The flap was sutured tension-free and the patient was dismissed. After the healing period, the patient noted improvement of the symptoms. One year later, a periapical radiography was taken showing the healing of the periapical radiolucency, the fixture showed no signs and symptoms of peri-implantitis, and tooth number 15 showed no mobility and no symptoms. Probing depth was 2 mm around the tooth and 3 mm around the implant (Figure 6D).

The histological analysis of the surrounding tissue revealed the presence of a rich inflammatory tissue.

## 4. Discussion

Among implant failures, a significant role seems to be played by apical peri-implantitis (IPL). Different researchers have described cases of IPL—as retrospective studies or case reports—with indications for treatment of this disease and suggestions for possible causes [8,9,11,12,13,14,15,16,17,18,19,20,21,22,23,24,25,26,27,28,29]. These include microfractures of the cortical bone, in the form of vestibular or lingual bony dehiscence and fenestration, osteitis or osteomyelitis secondary to bacterial contamination during the surgical procedure, overheating of the bony structure, or poor bone quality [7,8,9,11,19,20,21,22,23].

However, the best evidence-supported etiology of IPL suggested by the different researchers is bacterial in nature [8,11,12,19,22]. This etiology may be due either to transmission of endodontic periapical lesions, starting with dental structures adjacent to the implant, or to infected dental remnants present in the affected edentulous crest. If implants are inserted close to teeth with acute or chronic endodontic periapical lesions, or radicular fragments of extracted teeth remaining in the alveolus and not promptly eliminated, these can involve the surface implant and induce IPL [11,20,27,28,29].

From a statistical perspective, the region most affected by IPL is that of the maxillary premolars [8]. This may be due to the anatomy of these teeth, which often present with two roots. In many cases, incomplete flushing, shaping, and closure of the canal system can lead to failure of endodontic treatment and concomitant formation of apical inflammatory tissue, which in 26% of cases is not visible on radiography [24]. Extraction of the maxillary premolars and later revision of the alveolus with alveolar curettes can result in apical residues remaining permanently in the medullary bone if the procedures are not performed correctly. If not guided by precise radiographic examinations, substitution of the extracted structure with an implant in the infected zone can induce peri-implantitis in the apical area of the fixture. This occurrence can precipitate pain symptoms in the patient immediately after the intervention, as in our case, or even several weeks or years after the implantation [8,15,17,18,21,25,26].

IPL has been attributed in a histological study to the amide present on the latex of the gloves. In fact, the presence of amide from gloves has been detected inside neutrophilic granulocytes. However, this may not be the primary cause of IPL in the first case report, since the clinical symptomatology was manifested approximately one month after implant insertion and the adjacent dental structure presented an endodontic apical lesion [18].

Confirmation of the bacterial origin of apical IPL is supported by the finding of types of bacteria in the apical portion of the implant that are very similar to those in endodontic infections, such as Porphyromonas endodontalis, Porphyromonas gingivalis, and Prevotella intermedia, while contrariwise, bacteria typical of periodontal lesions are found in classic peri-implantitis [30,31,32,33,34,35,36,37,38].

Our study is one of the few available in the literature that reports an implant affected by apical IPL. The histological analysis revealed complete osteointegration of the coronal two-third of the fixture, whereas the apical portion remained free of contact to bone. This could be due to necrotic tooth remnants found upon CT scan examination. 

The osteotomy use of surgical burs or the fixture placement itself may have reactivated a latent infection around these tooth fragments, causing an inflammatory reaction in the apical portion of the implant. Our failure in the histological examination to find inflammatory tissue may be due to detachment or to its seclusion within the mandibular bone during drilling of the implant [8,9].

Other research has also reported the presence of root fragments—not detected during the implantation—which induced inflammatory apical reactions to the fixture [8].

Radiopaque zones in edentulous crests are very frequent in areas where a dental extraction has previously been performed. These may be due to focal osteosclerosis, a non-expansible radiopaque alteration of trabecular bone of unknown origin, asymptomatic, with various shapes and sizes, affecting both the maxilla and the mandible, with higher prevalence in the mandibular molar and premolar region. Its radiopacity may resemble other pathologies of the jaws, such as condensing osteitis, root segments, hypercementosis, cementoblastoma, impacted teeth, focal cemento-osseous dysplasia, and odontomas [39,40,41].

Focal osteosclerosis can be present in proximity to the retained root tips or in the edentulous crest region [42]. In our case, previous extraction of a dental structure may have resulted in focal osteosclerosis, possibly reactivated on the occasion of implant insertion in the area of interest. This suggestion would explain the area of radiopacity visible on both CT and intraoral radiography at teeth 46–47. As far as we know, however, there are no other published studies of implants with IPL in proximity to areas of focal osteosclerosis.

Another possible approach that would explain the patient’s painful symptomatology could be injury to the vascular-neural bundle of the alveolar canal occurring during preparation of the implant site. This may be seen also on radiography as an area of radiolucency involving the apex of the implant in close contact with the underlying alveolar canal. However, despite the fact that trunk anesthesia was not performed—to maintain neural sensitivity during the intervention—the patient did not experience any pain during placement of the implant or post-operative paresthesia. 

Treatment of local IPL consists in the removal of the implant (case report n.1) or removal of its apical portion (case report n.2), with consequent elimination of the inflammatory tissue [26]. Removal of the apical portion of the implant requires the remaining integrated part to be capable of bearing the masticatory load [9]. In case report n.1, we decided to remove the implant in toto since excision of its apical portion alone could have caused permanent injury to the vascular-neural bundle of the inferior alveolar nerve, with consequent neuro-sensory changes in the patient’s right lower lip. Instead, in case report n.2, the apical portion of the implant as well as the adjacent tooth apex were surgically removed and periapical lesion successfully healed with no signs and symptoms of IPL. In the first clinical case, the fixture was implanted in another surgical site; therefore, the healing times are crucial. However, it was a four-wall defect, which is a common post-extraction site. In the second case, the lesion had endodontic origin; thus, improvement was noticed 12 months after baseline.

Early surgical treatment is essential in order to limit progression of the lesion to the osseointegrated part of the fixture. Our histological study shows that this surgical therapy may be helpful in the treatment of IPL since the coronal two-thirds portion of the implant is fully osseointegrated. According to the present literature, some dental remnants may be intentionally left in post-extraction sites without causing infectious complications like the ones experienced by the authors in the first case report. This confirms that infections proceeding from neighboring teeth may cause IPL in fixtures’ periapical tissue. As far as investigated by the authors, only one failure for infectious reasons has been described with the socket shield technique because the apex was left during the surgical procedure. Indeed, the tooth apex and endodontium need to be accurately removed when performing this technique in order to achieve clinical success; only a buccal sliver is meant to be left in situ [43,44].

## 5. Conclusions

Because of the ever-increasing number of people undergoing implant-based treatment in recent years, there has been an increased incidence of apical IPL. The presence of infected root remnants or endodontic periapical lesions seems to be the major cause of this type of implant failure. Treatment of apical IPL should be directed toward elimination of the bacterial noxae and limitation of progression of the lesion to the osseointegrated part of the implant. Removal of the apical portion of the implant appears to be a valid therapeutic option, since the remaining part of the fixture appears from our histological examination to be optimally osseointegrated and not affected by the inflammatory process. Good initial treatment planning seems fundamentally important. It should be targeted toward the identification of local etiological factors present either in the edentulous crest or the adjacent dental structures, and their preventive elimination before insertion of the fixture so as not to compromise implant predictability.

## Figures and Tables

**Figure 1 bioengineering-09-00145-f001:**
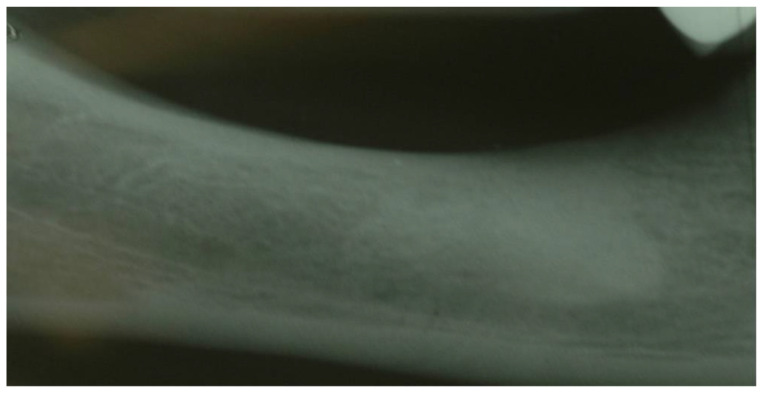
Preoperative intraoral radiograph of the edentulous area.

**Figure 2 bioengineering-09-00145-f002:**
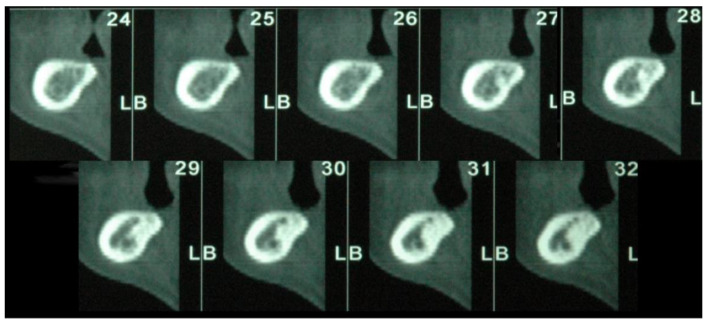
Preoperative CT scan of #45–47 region: an ill-defined radiopacity is visible, which could be due to the presence of a root fragment left in the alveolus at the time of extraction of #46.

**Figure 3 bioengineering-09-00145-f003:**
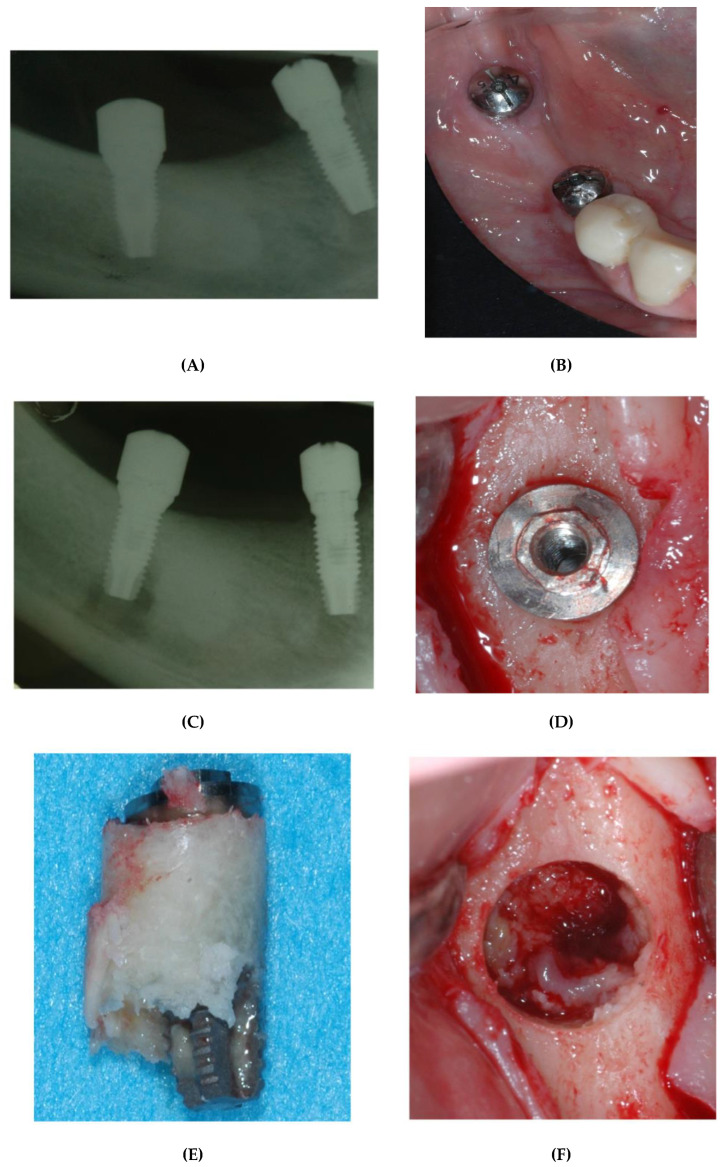
(**A**) Postoperative intraoral radiograph showing the position of the implants at #45 and #47. (**B**) Intraoral radiograph taken one month after the intervention. An area of radiolucency is visible at the apex of the implant at #47. (**C**) Clinical image 6 weeks after the implantation: there are no visible clinical signs of inflammation in the oral mucosa. (**D**) Clinical image of the implant at #47 after a mucoperiosteal flap was raised. The implant appears perfectly integrated and does not seem to have undergone crestal resorption. (**E**) The implant shortly after removal. The apical portion of the implant is not in contact with the bone. (**F**) Bony crest at #47 after removal of the implant. Note the presence of granulation tissue at the bottom of the cavity.

**Figure 4 bioengineering-09-00145-f004:**
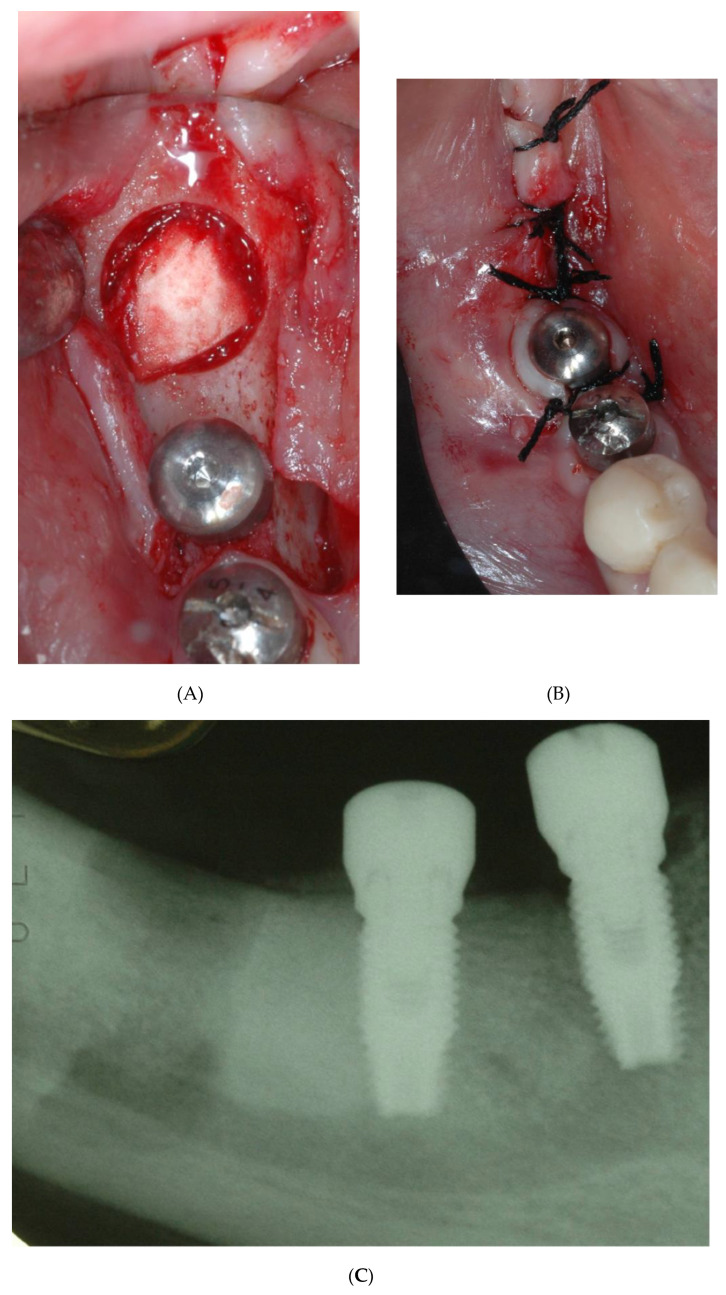
(**A**) Same area after packing with a collagen sponge and insertion of an implant at #46. (**B**) Clinical image after insertion of a healing screw in the implant at #46 and initial closure with 4/0 interrupted silk sutures. (**C**) Postoperative intraoral radiograph of area #45–#47. (**D**) Radiograph one year after the definitive prosthetic restoration.

**Figure 5 bioengineering-09-00145-f005:**
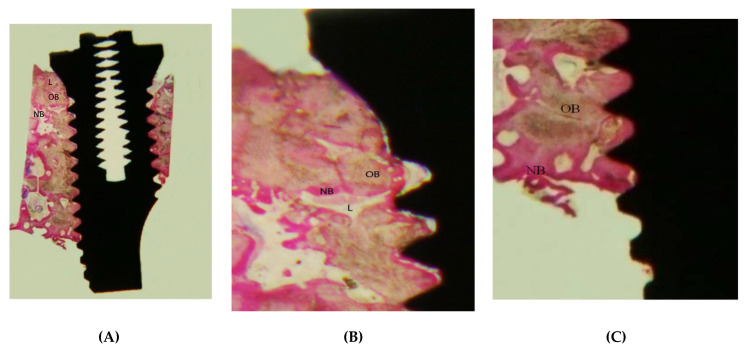
(**A**) In the coronal segment of the implant the old bone (OB) is in contact with the trabeculae of new bone (NB), interspersed with bony lacunae (L). The detachment of the first three threads of the implant from the bone seems to be due to a histological preparation artifact (basic fuchsin, 15× magnification). (**B**) Direct contact of the old bone (OB) and new bone (NB) with the implant thread in the apical portion of the implant (basic fuchsin, 15× magnification), L = lacunae. (**C**) Histological view of the removed implant. Only the middle and coronal segments of the implant are in contact with bone (1.5× magnification).

**Figure 6 bioengineering-09-00145-f006:**
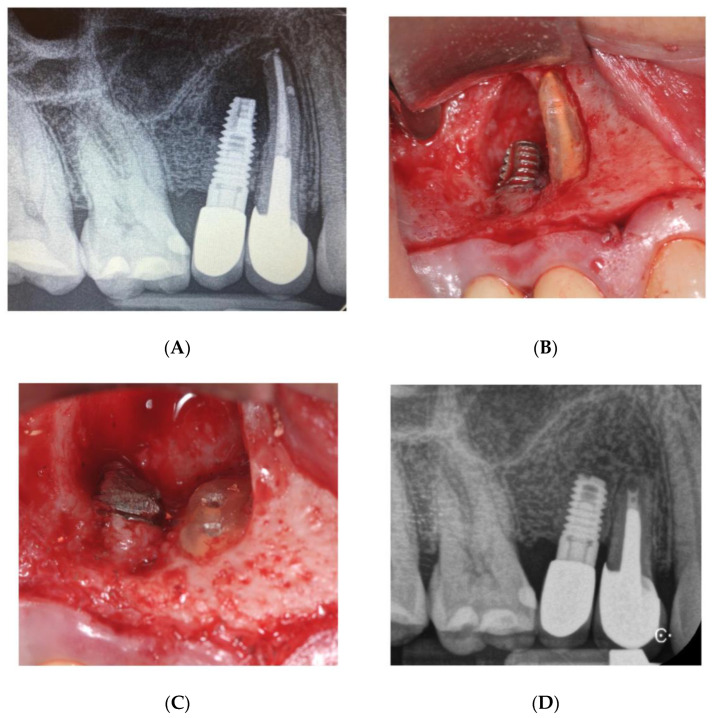
Case 2 (**A**) Preoperative image showing the presence of a radiolucency between the implant apex and the adjacent tooth. (**B**) Clinical view of the apex prior to the resection. (**C**) Implant apical resection and tooth apex resection with both root canals ready to receive the filling material. (**D**) 2 years follow-up.

## Data Availability

The data supporting the findings of the present study are available from the corresponding author upon reasonable request.

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
