# Peer review of "Implant Periapical Lesion: Clinical and Histological Analysis of Two Case Reports Carried Out with Two Different Approaches"

_bioengineering, 2022, doi:10.3390/bioengineering9040145_

Round 1
Reviewer 1 Report
Dear authors,
thank you for the opportunity to review this interesting paper.
General: It is a nice presentation of two cases with synoptic clinical, radiographic and histological analysis and therefore very valuable for clinical researchers and practicioners. However, there are some points that need to be adapted. These points are listed below.
Ethics: You must add the information that the patients provided informed consent for using their informations and images.
Title: Please use a grammatically more clear title.
L. 42: Periodontitis and Periimplantitis as non-comminicable diseaes with possible interactions between host immune denfense and bacteria should also be named as common reasons for implant failures.
L. 84: To adequatly name the teeth, the FDI numbering system should be used in the manuscript.
L. 86: Why were prophylactic antibiotics prescribed? Did the patient show any general diseases (diabetes, immunodeficiency etc.)?
For a case report dealing with implants showing apical lesions it would be important to give an information about any periapical lesions of former teeth from the anamnesis.
L. 96: Was the implantation performed fully or partailly guided?
L. 114: Please give a definition of "excessive probing depth". In general, exact Probing depths for all time points should be listed.
Case 2: Also for this case, you should add information about any endodontic problems regarding former 15 before extratction.
Can you provide histological informations as in case 1?
L. 213: Missing interpunctation. Please check the whole manuscript.
L. 215-216: Please add information acout tooth mobility, pocket depth and other clinical parameters at 12 months.
L. 275: Missing reference
L. 276: Please be more specific about the reasons of focal osteosclerosis. In the liteature, many local and systemic factors are discussed. Please add.
L. 292-294: Regrading the treatment of IPL, please be more specific about the healing times before renewed implantation or until complete bone regeneration in the discussion and the case reports.
Discussion general: You discussed residual tooth parts as a possible reason of IPL. There are some practicioners that use and recommend techniques that intentionally leave tooth parts, such as tissue master concept (P. neumeyer) or socket shield technique.
Please include this point and accordingly some references in your discussion.
L. 311: As bacteria were not investigated in your study, you should place it not that prominently in your conclusion.
Author Response
Dear Reviewer, in this letter i answer point by point your comments:
Ethics: You must add the information that the patients provided informed consent for using their informations and images.
-Informed consent was given to both patients and It was signed down prior to the procedure. Patients were fully aware of the pathological condition they were suffering from and before the surgery all the alternatives were discussed with them. Thus, a specific informed consent was written and signed down by the patients.
Title: Please use a grammatically more clear title.
-Implant Periapical Lesion: two case reports carried out with two different approaches. Clinical and Histological analysis.
L. 42: Periodontitis and Periimplantitis as non-comminicable diseaes with possible interactions between host immune denfense and bacteria should also be named as common reasons for implant failures.
-Periodontitis and Periimplaintitis may trigger interactions between host immune defense mechanism and bacteria that eventually lead to implant failure. (I inserted this sentence in the text).
L. 84: To adequatly name the teeth, the FDI numbering system should be used in the manuscript.
-Done.
L. 86: Why were prophylactic antibiotics prescribed? Did the patient show any general diseases (diabetes, immunodeficiency etc.)?
-For a case report dealing with implants showing apical lesions it would be important to give an information about any periapical lesions of former teeth from the anamnesis.
We made a mistake: we meant to write antibiotic therapy instead of antibiotic prophylaxis.
L. 96: Was the implantation performed fully or partailly guided?
-No it was performed free-handedly
L. 114: Please give a definition of "excessive probing depth". In general, exact Probing depths for all time points should be listed.
-The probing depth was 3 mm.
Case 2: Also for this case, you should add information about any endodontic problems regarding former 15 before extratction.
Can you provide histological informations as in case 1?
-The endodontic therapy seemed to be correctly performed, the obturation material was throughly compacted, lateral accessory canals were present and filled. The tooth was restored with a metallic post and core and crown.
The authors sent the specimen to the lab for further analysis but unfortunately they only received the report and no pictures.
L. 213: Missing interpunctation. Please check the whole manuscript.
-Checked
L. 215-216: Please add information about tooth mobility, pocket depth and other clinical parameters at 12 months.
-After a 12-months follow-up the gingiva was healthy, teeth were sound and there were no signs of mobility. Probing depth was 2 mm around the tooth and 3 mm around the implant.
L. 275: Missing reference
-We added the missing reference.
L. 276: Please be more specific about the reasons of focal osteosclerosis. In the liteature, many local and systemic factors are discussed. Please add.
-Done
L. 292-294: Regrading the treatment of IPL, please be more specific about the healing times before renewed implantation or until complete bone regeneration in the discussion and the case reports.
-In the first case the fixture was implanted into another surgical site, therefore the healing times are crucial. However, it was a 4-walls defect as a common post-extraction site.
In the second case the lesion had endodontic origin, thus improvement was noticed after 1 year from baseline
Discussion general: You discussed residual tooth parts as a possible reason of IPL. There are some practicioners that use and recommend techniques that intentionally leave tooth parts, such as tissue master concept (P. neumeyer) or socket shield technique.
-The endodontium is eliminated when dental remnants are intentionally left in the surgical site. Instead, in these case reports the lesion had endodontic origin.
Please include this point and accordingly some references in your discussion.
-Done
L. 311: As bacteria were not investigated in your study, you should place it not that prominently in your conclusion.
Done
Reviewer 2 Report
Some should be improved before considering publication
- histologic analysis: I think there are no valuable information from the histology. There is no tissue content in the apical area of the implant.
- photos: please consider to leave only the pictures pertinent to IPL.
- I doubt the second case was really true IPL. It is more likely to develop the lesion from the tooth site. In the existing literature, was such kind of the lesion called IPL?
- line 107 - 110: is it right to write that patient reported IPL? IPL may not be the term the patient used. The patients may report pain or unusual sensation. IPL is the authors' diagnosis.
- lines 252- 253: when was the implant in the first case placed? please check (3.5 years?). And can the amide of the glove contaminate the implant surface? Generally, implant surface is not touched with the glove.
- Did you consider heating during drilling procedure in the first case?
- "Discussion" part should improved. The current one is not organized.
Author Response
Dear Revisor I answer to your comments point by point.
- histologic analysis: I think there are no valuable information from the histology. There is no tissue content in the apical area of the implant.
The histology shows no valuable information because the granulation tissue was non attached to the implant immediately after its carotation, but instead it remained in the surgical site and it was just removed. The histology clearly shows that there had been no osteointegration in the apical portion of the implant.
- photos: please consider to leave only the pictures pertinent to IPL.
In our opinion, the photos are a good representation of the clinical cases. Please point out which pictures you would like to remove, if you like.
- I doubt the second case was really true IPL. It is more likely to develop the lesion from the tooth site. In the existing literature, was such kind of the lesion called IPL? Per IPL
IPL is intended as a pathological condition in which only the apical third of a fixture loses osteointegration, whereas the coronal portion remains osteointegrated. Several cases of IPL are described in the existing literature and a periapical lesion of a neighboring tooth can be considered a IPL. Here you have some bibliography about it:
.Zhou W, Han C, Li D, Li Y, Song Y, Zhao Y. Endodontic treatment of teeth induces retrograde peri-implantitis. Clin Oral Implants Res. 2009;20:1326-32.
Temmerman A, Lefever D, Teughels W, Balshi TJ, Balshi SF, Qui- rynen M. Etiology and treatment of periapical lesions around dental implants. Periodontol 2000. 2014;66:247-54.
Brisman DL, Brisman AS, Moses MS. Implant failures associated with asymptomatic endodontically treated teeth. J Am Dent Assoc. 2001;132:191-5.
Lefever D, Van Assche N, Temmerman A, Teughels W, Qui- rynen M. Aetiology, microbiology and therapy of periapical lesions around oral implants: a retrospective analysis. J Clin Periodontol. 2013;40:296-302.
Blaya-Tárraga, JA, Cervera-Ballester, J, Peñarrocha-Oltra D, Peñarrocha- Diago M. Periapical implant lesion: a systematic review. Med Oral Patol Oral Cir Bucal. Med Oral Patol Oral Cir Bucal. 2017 Nov 1;22 (6):e737-49.
- line 107 - 110: is it right to write that patient reported IPL? IPL may not be the term the patient used. The patients may report pain or unusual sensation. IPL is the authors' diagnosis.
We made the corrections as you suggested. The patient reported continuous pain, similar to pulpits, after the implant surgery. The authors diagnosed IPL after taking intraoral radiographs.
- lines 252- 253: when was the implant in the first case placed? please check (3.5 years?). And can the amide of the glove contaminate the implant surface? Generally, implant surface is not touched with the glove.
It was a mistake, the implant was placed around 30 days later. Sometimes the implant carrier accidentally does not engage the implant head correctly and it needs to be manually engaged. This maneuver should be done with titanium pliers ma sometimes clinicians do not have them and they touch the implant with their gloves. This is when contamination occurs.
- Did you consider heating during drilling procedure in the first case?
It could be one of the causes leading to IPL, but in the first clinical case, upon CT examination, a root fragment is evident.
- "Discussion" part should improved. The current one is not organized. We added some reference and some crucial point as suggested by your co-reviewer. I hope this discussion is better than the first one.
Thank you for your work.
Round 2
Reviewer 1 Report
Dear authors,
thank you for the renewed opportunity to review the interesting manuscript. It has improved significantly.
Please add the information to the manuscript, that all implantations were placed free-handed.
Author Response
Dear Reviewer .
I add that the implant were inserted free-handed.
thank you
Reviewer 2 Report
- please refer to other articles for organizing the photos. I recommend to prepare multi-panel one.
- why did you mention socket shield technique in the discussion? That is not relevant to your cases.
- Histologic section just corresponded to the finding in radiographs. Nothing more important finding was given. I think you just mention about a lack of osseo-integration in the apical part of the implant. Bone to implant contact is not relevant. Considering the above, re-write the result and discussion.
- lines 304- 305: 3.5 years?. this is still visible.
- case2 -> did you consider the effect of sealer material leak?
Author Response
Dear Reviewer, I answer step by step to your questions.
- please refer to other articles for organizing the photos. I recommend to prepare multi-panel one.
We tried to remove the images and make a multiplanel for each individual case, if you prefer this type of graphics we will delete all the calls relating to the missing images in the text and insert the correct citations.
- why did you mention socket shield technique in the discussion? That is not relevant to your cases.
It was a request from Reviewer 1: during the first review he asked us to explain why a modern technique such as the socket shield technique does not lead to IPL and we answered to this request by reporting it in the text.
- Histologic section just corresponded to the finding in radiographs. Nothing more important finding was given. I think you just mention about a lack of osseo-integration in the apical part of the implant. Bone to implant contact is not relevant. Considering the above, re-write the result and discussion.
We made the corrections you suggested
- lines 304- 305: 3.5 years?. this is still visible.
It was a mistake. We made the corrections.
- case2 -> did you consider the effect of sealer material leak?
It could be the cause. We have no information regarding the previous endodontic treatment. Although the length and compaction of the material seem adequate, the shaping does not have an adequate taper and probably the anatomy of the apical third has been altered, perhaps with the creation of a large (> 50) and teardrop foramen that is difficult to obturate with conventional techniques and materials.
